# Fermentation of *Betaphycus gelatinum* Using *Lactobacillus brevis*: Growth of Probiotics, Total Polyphenol Content, Polyphenol Profile, and Antioxidant Capacity

**DOI:** 10.3390/foods12183334

**Published:** 2023-09-06

**Authors:** Zhe Wang, Caibo Zhao, Zhiqiang Guo, Shuyi Li, Zhenzhou Zhu, Nabil Grimi, Juan Xiao

**Affiliations:** 1Hainan Engineering Research Center of Aquatic Resources Efficient Utilization in South China Sea, Key Laboratory of Seafood Processing of Haikou School of Food Science and Engineering, Hainan University, Haikou 570228, China; 2School of Marine Science and Engineering, Hainan University, Haikou 570228, China; 3National R&D Center for Se-Rich Agricultural Products Processing, Hubei Engineering Research Center for Deep Processing of Green Se-Rich Agricultural Products, School of Modern Industry for Selenium Science and Engineering, Wuhan Polytechnic University, Wuhan 430023, China; 4Centre de Recherche Royallieu, Université de Technologie de Compiègne, Sorbonne Universités, CS 60319, 60203 Compiègne CEDEX, France

**Keywords:** *Betaphycus gelatinum*, *Lactobacillus brevis*, polyphenol profile, antioxidant activity

## Abstract

Little information is available regarding polyphenol variations in the food processing of edible and medicinal red seaweed, *Betaphycus gelatinum*. This study investigated the effects of *Lactobacillus brevis* fermentation on total polyphenol content (TPC), polyphenol profile, and antioxidant activity in *Betaphycus gelatinum* pretreated by ultrasound-assisted mild acid hydrolysis for the first time. During 60 h of fermentation, the viable colony number significantly increased, pH significantly decreased, and reducing sugar content significantly decreased initially, then significantly increased. Free TPC significantly increased to 865.42 ± 29.29 μg GAE/g DW (163.09% increase) with increasing antioxidant activity, while bound TPC significantly decreased to 1004.90 ± 87.32 μg GAE/g DW (27.69% decrease) with decreasing antioxidant activity. Furthermore, 27 polyphenol compounds were identified by ultra-high-performance liquid chromatography with Xevo triple quadrupole mass spectrometry. In total, 19 and 23 free polyphenols and 24 and 20 bound polyphenols were identified before and after fermentation, respectively. Before fermentation, bound *trans*-cinnamic acid (56.75%), bound rosmarinic acid (26.62%), and free *trans*-cinnamic acid (3.85%) were the main components. After fermentation, free rosmarinic acid (43.57%), bound *trans*-cinnamic acid (15.19%), bound rosmarinic acid (13.33%), and free *trans*-cinnamic acid (5.99%) were the main components. These results provide information for the food processing of *Betaphycus gelatinum*.

## 1. Introduction

*Betaphycus gelatinum*, an edible seaweed belonging to the genus *Betaphycus*, family *Solieriaceae,* and phylum *Rhodophyta*, has been utilized in local food and traditional Chinese medicine for centuries [1]. It is rich in carrageenan, dietary fiber, and minerals and low in fat and calories; thus, it is considered to have high nutritional and health value. As one of the main cultivated seaweeds in Hainan, *Betaphycus gelatinum* is usually used as a raw material for the extraction of carrageenan [2], but not for food processing. An increasing number of studies are focusing on the gel characteristics of carrageenan from *Betaphycus gelatinum* [2,3]. However, few studies have focused on the variation in nutrients and phytochemicals, including polyphenols, in the food processing of *Betaphycus gelatinum*. Seaweed species *Kappaphycus* and *Eucheuma* (belonging to the family *Solieriaceae*) are rich in polyphenols with antioxidant, tumor-suppressive, and anti-estrogenic activities [4,5,6], indicating that *Betaphycus gelatinum* may be rich in polyphenols with health-promoting effects. However, to the best of our knowledge, polyphenols from *Betaphycus gelatinum* have not been well studied to date.

Polyphenols, widely found in fruits, vegetables, grains, and seaweeds, are generally present in free and bound forms [7]. Seaweed is rich in polyphenols, including phloroglucinol, flavonoid, bromophenol, phenolic acid, and proanthocyanidins, and has a high proportion of bound forms [8,9]. Free polyphenols are usually found in plant vacuoles, while bound polyphenols are usually linked to macromolecular polysaccharides such as pectin and cellulose [7]. Compared to free polyphenols, bound polyphenols are difficult for the small intestine to absorb and utilize directly [10]. The unabsorbed polyphenols are usually released into free polyphenols and then biotransformed by microorganisms in the colon; meanwhile, the microbial metabolites of polyphenols promote the growth and proliferation of microorganisms, suggesting that in vitro lactic acid bacteria fermentation may change the polyphenol profile [7,11].

Lactic acid bacteria fermentation is a common food processing technology for vegetables, grains, dairy products, and fruits [12,13,14]. During fermentation, lactic acid bacteria produce enzymes to degrade macromolecular carbohydrates into micromolecular sugars and functional oligosaccharides, which promote the growth of lactic acid bacteria. Moreover, many beneficial metabolites, such as lactic acid and short-chain fatty acids, are produced, which improve the nutritional values and are beneficial for food preservation [12,13,14,15]. Moreover, lactic acid bacteria fermentation has positive impacts on sensory quality, thus improving consumer acceptance [13,15]. Importantly, fermentation by microorganisms has been reported to play an important role in metabolizing phytochemicals, including polyphenols [13,15]. Usually, the enzymes produced by lactic acid bacteria can break down the linkages between polyphenols and polysaccharides to release the bound polyphenols in their free form and contribute to polyphenol conversion [16]. The process improves the bioavailability and physiological functions of polyphenols [13,14,15].

Several studies have focused on the lactic acid bacteria fermentation of seaweeds [17,18]. The *Lactobacillus plantarum* fermentation of three edible Irish brown seaweeds promoted an increase in the antioxidant activity of the fermentation broth [17]. Lactic acid bacteria isolated from a marine source were used to ferment *Sargassum* sp., with significant decreases in pH value and reduced sugar content and significant increases in culture viability, total acid content, and anticoagulant and antioxidant activities [18]. Three types of *Lactobacillus plantarum* strains isolated from the coast decreased the pH and increased O_2_^−^ radical-scavenging capacities without a significant effect on the total polyphenol content (TPC) in free extracts of *Gloiopeltis furcata*, *Chondrus elatus*, and *Monostroma nitidum*, but could not ferment the other eight seaweeds [19]. The brown alga *Sargassum siliquastrum* was fermented by isolated lactic acid bacteria, including *Weissella* sp., *Lactobacillus* sp., *Leuconostoc* sp., and *Streptococcus* sp., with increased free TPC, total flavonoid content, DPPH radical scavenging activities, and/or angiotensin converting enzyme inhibitory activity [20]. Free antioxidant activity and the TPC of *Arthrospira platensis* increased significantly after fermentation by *Lactobacillus plantarum* [21]. Although these results indicated that lactic acid bacteria fermentation can change free TPC and antioxidant ability, its effects on the free and bound polyphenol profiles of seaweeds have not been well studied. Moreover, there is no available information about the effect of fermentation by lactic acid bacteria on the variations of free and bound polyphenols in *Betaphycus gelatinum.*

*Lactobacillus brevis* was identified as one of the main microbes in naturally fermented Samjung-hwan (a traditional fermented herb formula), in which increased TPC, total flavonoid content (TFC), and antioxidant ability were found when compared with unfermented Samjung-hwan [22]. Fermentation by *L. brevis*, isolated from maize and sorghum, improved the tannin content and TPC of maize flour with an increasing fermentation period [23]. *L. brevis* fermentation increased the TPC, TFC, γ-aminobutyric acid content, antioxidant ability, and angiotensin-converting enzyme inhibitory activity in Chinese yam [24]. An edible brown seaweed, *Saccharina japonica*, as the sole carbohydrate source, was fermented by *L. brevis* [25]. The results of this study indicated that *L. brevis* may be an effective microbe for the fermentation of *Betaphycus gelatinum*.

In the present study, *Betaphycus gelatinum* pretreated by ultrasound-assisted mild acid hydrolysis, according to the methods of our previous study, was fermented by *L. brevis* for the first time. The dynamic growth of *L. brevis* and physicochemical indexes were monitored. Secondly, TPC, the polyphenol profile, and antioxidant activity in both free and bound polyphenol extracts after fermentation for 0 and 60 h were measured. Finally, the correlation between free and bound TPC and antioxidant activity was analyzed. The results of this study provide useful information for the food processing of *Betaphycus gelatinum*.

## 2. Materials and Methods

### 2.1. Materials

The fresh *Betaphycus gelatinum* was provided by the Hainan Academy of Marine and Fisheries Sciences. After soaking in tap water for 12 h for desalination, *Betaphycus gelatinum* was freeze-dried by a freeze-dryer (DGJ-25C, Shanghai Boden Biotechnology Co., Ltd., Shanghai, China), crushed into a 40-mesh size, packed, and stored at −20 °C.

### 2.2. Chemicals and Reagents

Formic acid, acetonitrile, and methanol for UPLC-QQQ-MS were purchased from Sigma Chemical Co., Ltd. (Shanghai, China). Cyanidin, 1,3-O-dicaffeoylquinic acid, epicatechin gallate, ferulic acid, 4-hydroxybenzoic acid, isorhamnetin-3-O-glucoside, myricetin, quercetin, taxifolin-7-O-rhamnoside, and vitexin for UPLC-QQQ-MS were purchased from Qiyun Biotechnology (Guangzhou, China). Caffeic acid, cosemetin, 2,4-dibromophenol, ellagic acid, epigallocatechin, ethyl vanillin, gallocatechin, hinokiflavone, hyperoside, isoquercitrin, kaempferol-3-O-glucoside, morin, myricitrin, rosmarinic acid, taxifolin, *trans*-cinnamic acid, and 2,4,6-tribromophenol for UPLC-QQQ-MS were purchased from Yuanye Biotechnology Co., Ltd. (Shanghai, China). Total antioxidant capacity assay kits with the ferric-reducing antioxidant power (FRAP) assay and 2,2′-azino-bis(3-ethylbenzthiazoline)-6-sulfonic acid (ABTS) radical scavenging activity assay were purchased from the Nanjing Jiancheng Bioengineering Institute (Nanjing, China). D-Galactose (analytical grade) and 1,1-diphenyl-2-picrylhydrazyl (DPPH) were purchased from Macklin Biochemical Technology Co., Ltd. (Shanghai, China). De Man, Rogosa, and Sharpe (MRS) medium and broth were purchased from the Guangdong Huankai Microbial Technology Co., Ltd. (Guangzhou, China), and 3,5-dinitrosalicylic acid (DNS) was purchased from the Beijing Solarbio Technology Co., Ltd. (Beijing, China). All other chemicals and reagents were of analytical grade.

### 2.3. Preparation of the Betaphycus Gelatinum Substrate for Fermentation

*Betaphycus gelatinum* was pretreated by ultrasound-assisted mild acid hydrolysis to prepare the substrate for fermentation according to the methods of our previous study [26]. Briefly, 1.25 g of *Betaphycus gelatinum* powder was mixed with 50 mL of distilled water and 0.4 mL of HCl (6 mol/L). The mixture was mixed and placed in a bath-type ultrasonic instrument (SB25-12DTD, Ningbo Xinzhi Biotechnology Co., Ltd., Ningbo, China) at 75 °C under an ultrasonic power of 612 W for 10 min. After being rapidly cooled, the mixture was adjusted to pH 6.8–7.0 and then sterilized by a pressure steam sterilizer (YXQ-LB-50SII, Shanghai Boxun Industrial Co., Ltd., Shanghai, China) at 121 °C for 15 min.

### 2.4. Preparation of the L. brevis Starter

For the preparation of the starter, 10 mL of sterilized MRS broth was inoculated with 0.1 mL of *L. brevis* stock culture (Beijing Microbiological Culture Collection Center, Beijing, China) and incubated at 37 °C for 12–24 h in a thermostatic shaker (ZQTY-70N, Shanghai Zhichu Instrument Co., Ltd., Shanghai, China) to obtain a working starter containing 6–7 log CFU/mL cells. After the working starter was centrifuged by a high-speed freezing centrifuge (H2050R, Hunan Xiangyi Laboratory Instrument Development Co., Ltd., Changsha, China), the precipitate was suspended using sterilized normal saline to obtain the *L. brevis* starter.

### 2.5. Fermentation

The *Betaphycus gelatinum* substrate was inoculated with 1% (*v*/*v*) of *L. brevis* starter, which was then incubated in a biochemical incubator (LRH-250, Shanghai Yiheng Scientific instrument Co., Ltd., Shanghai, China) at 37 °C for 60 h. Samples were taken at 0, 12, 24, 36, 48, and 60 h, respectively, for microbiological and chemical analyses.

### 2.6. Determination of the Viable Colony Number, pH, and Reducing Sugar Content

The pH of the fermentation broth was measured using a pH meter (FE28, Mettler-Toledo Technology (China) Co., Ltd., Shanghai, China). The viable *L. brevis* colony number in the fermentation broth was determined by the standard plate count method and expressed as log CFU/mL [13]. In brief, the diluted fermentation broth was coated on an MRS agar plate, cultured at 37 °C, and the colonies were counted. The reducing sugar content was determined using the DNS method with D-galactose as the standard [27]. Briefly, the diluted fermentation broth was placed in a 25 mL colorimetric tube, followed by the addition of 1.5 mL of DNS solution, and the mixture was then incubated at 100 °C for 10 min. The mixture was cooled to 25 °C with ice-cold water and filled to the 25 mL mark with distilled water before testing. The absorbance was measured at 520 nm by an ultraviolet-visible spectrophotometer (UV-5100, Shanghai Yuanxi Instrument Co., Ltd., Shanghai, China). The reducing sugar content was expressed as mg/g dry weight (DW) of seaweed powder (mg/g DW).

### 2.7. Extraction of Free and Bound Polyphenols

The free polyphenol extracts were obtained as described previously [28], with some modifications. Briefly, the *Betaphycus gelatinum* substrate without/with fermentation for 60 h was centrifuged at 8000 rpm for 10 min at 4 °C, and the supernatant was extracted with ethyl acetate six times. The ethyl acetate fractions were mixed and concentrated to dryness on a rotating evaporator (RE212-B, Yamatuo Technology Trading Co., Ltd., Chongqing, China) at 45 °C, then redissolved in 5 mL of 85% methanol and stored at −18 °C.

The bound polyphenol extracts were obtained from the precipitate separated from the fermentation broth using the acid hydrolysis method described previously [9] with some modifications. Briefly, the precipitate was hydrolyzed with HCl (40 mL, 6 mol/L) for 24 h with continuous shaking under nitrogen gas and then centrifuged at 8000 rpm for 10 min at 4 °C. After the extraction was repeated, the combined supernatants were adjusted to pH 6.8–7.0 and then extracted with ethyl acetate six times. The ethyl acetate fractions were concentrated to dryness at 45 °C, then reconstituted in 5 mL of 85% methanol and stored at −18 °C.

### 2.8. Determination of Total Phenolic Content (TPC)

The TPC of free and bound polyphenol extracts was determined using the Folin–Ciocalteu colorimetric method with gallic acid as the standard [9]. In short, the diluted extract was mixed with Folin–Ciocalteu reagent and incubated. After 6 min, Na_2_CO_3_ solution (7%, *m*/*v*) was added. The samples were incubated in the dark for 1 h. Subsequently, the absorbance was determined at 760 nm using a microplate reader (RT-6100, Rayto Life and Analytical Sciences Co., Ltd., Shenzhen, China). The TPC was represented as μg gallic acid equivalent/g DW of seaweed (μg GAE/g DW). The sum of the TPC of free and bound polyphenol extracts was also calculated.

### 2.9. Polyphenol Profile

The polyphenol profiles of free and bound polyphenol extracts were obtained by the qualitative and quantitative analysis of individual polyphenol compounds on an ultra-high-performance liquid chromatograph with the Xevo triple quadrupole mass spectrometry system (UPLC-QQQ-MS, Waters, Milford, MA, USA) according to our previous methods [9]. The separation was performed on a Waters BEH-C18 column (1.7 μm, 2.1 i.d. × 100 mm) with 0.5% formic acid-water (elution A) and acetonitrile (elution B) at the following gradient program: 0~1 min 5% B; 8 min 25% B; 12 min 60% B; 13~16 min 100% B; 16.1~20 min 5% B. With a multiple reaction monitoring approach, the preliminary qualitative analysis of polyphenols was achieved by comparing the observed formula mass of parent and fragment ions with published values. According to the preliminary qualitative analysis, the basic structure of some peaks was deduced, and the related commercial standards were purchased. Then, further assignments were achieved using UPLC-QQQ-MS by comparing the retention time, formula mass, and mass spectra (MS) data of the polyphenol standards. The quantitative analysis was achieved through calculations with the calibration curve (Appendix A) of respective standards (μg/g DW). The mass range (100~1000), capillary voltage (5.5 kV for negative mode, 4.5 kV for positive mode), drying gas (N_2_) temperature (500 °C), and drying gas flow (1000 L/h) were set.

### 2.10. Antioxidant Ability

The antioxidant abilities of free and bound polyphenol extracts were determined by FRAP and ABTS radical scavenging activity assays using commercial kits as per the manufacturer’s protocols. The absorbance was determined at 593 and 734 nm, respectively. The DPPH radical scavenging capacity was measured using the method reported previously [29], and the absorbance was determined at 517 nm. The FRAP value was expressed as mmol ferrous sulfate equivalent/g DW of seaweed (mM Fe(II)E/g DW). The ABTS and DPPH radical scavenging capacities (%) were calculated using Equation (1):(1)$$Scavenging\;capacity\left(\%\right)=\frac{A_{io}-\left(A_{i}-A_{0}\right)}{A_{io}}\times 100\%$$
where Aio and Ai represent the absorbance of the reaction without and with the polyphenol extracts, respectively. A0 represents the absorbance of the polyphenol extracts with a working solution replaced by deionized water.

### 2.11. Statistical Analyses

The data are shown as the mean ± standard deviation (n = 3). The data were evaluated by one-way ANOVA, followed by Duncan’s multiple comparisons at a 0.05 probability level using SPSS v. 23.0. Correlation analysis was conducted using Pearson’s correlation tests following two-tail tests in SPSS v. 23.0, and the heatmap was constructed using Origin 2021 v. 9.8.0.200 software.

## 3. Results and Discussion

### 3.1. Viable Colony Number, pH, and Reducing Sugar Content

The changes in viable colony number, pH, and reducing sugar content during *L. brevis* fermentation are shown in Figure 1. During 0~12 h of fermentation, the viable colony number presented a minor increase from 5.61 ± 0.01 log CFU/mL to 5.79 ± 0.07 log CFU/mL (*p* < 0.05), which may have arisen from the adaptation of *L. brevis* to a new environment [17]. During fermentation from 12 to 24 h, the viable colony number increased to 6.93 ± 0.10 log CFU/mL (*p* < 0.05), with a 19.70% increase. The viable colony number increased slowly during 24~60 h, indicating that *L. brevis* was in the stationary phase. The viable colony number reached the maximum (7.09 ± 0.01 log CFU/mL) at 60 h, which conformed to the recommended level of probiotics in food at the time of consumption [30]. These results indicated that *Betaphycus gelatinum* can be fermented by *L. brevis*.

The initial pH value of the *Betaphycus gelatinum* fermentation broth was 5.35 ± 0.09 (0 h). Subsequently, a significant pH decrease was observed during 0~24 h (*p* < 0.05), and finally, the pH reached the lowest value (4.75 ± 0.02) at 60 h. During the fermentation process of *Saccharina japonica* by *L. brevis*, a quickly descending pH was initially observed, followed by a slow decrease in pH [25]. The changing trend of pH during fermentation is similar to that observed in this study. Lactic acid bacteria fermentation was indeed capable of producing organic acids and fatty acids to lower the pH, contributing to the inhibition of pathogenic bacteria proliferation and growth and the extension of food shelf life [13,14].

The reducing sugar content decreased from 120.73 ± 1.82 mg/g DW to the minimum (113.64 ± 1.31 mg/g DW) at 24 h. Subsequently, it increased significantly with the prolongation of fermentation time and reached its maximum (133.34 ± 0.97 mg/g DW) at 60 h (*p* < 0.05). The maximum at 60 h was higher than the initial value at 0 h (*p* < 0.05). It is well known that sugar is an important carbon source for lactic acid bacteria during fermentation [14,27,31]. In this study, *Betaphycus gelatinum* substrate was the sole carbon source for the proliferation, growth, and metabolism of *L. brevis*. Many studies have found that the level of reducing sugar or total sugar rapidly declined initially during fermentation and then slowly declined [14,15]. Usually, lactic acid bacteria can directly use the reducing sugar in the fermentation broth and produce enzymes to degrade macromolecular carbohydrates into micromolecular sugars, including reducing sugars [27,31]. The reducing sugar is altered dynamically during fermentation depending on the consumption rate of the existing reducing sugar and the production rate of the reducing sugar [27,31,32]. The reducing sugar content in the fermentation system of *Holothuria leucospilota* polysaccharides with fecal flora in vitro increased first and then decreased gradually [32]. A similar trend was also observed in the total sugar content of barley juice during *Bifidobacterium infantis* fermentation [29]. In this study, *Betaphycus gelatinum* pretreated with ultrasound-assisted mild acid contained reducing sugars, which were directly used by *L. brevis.* Meanwhile, *L. brevis* produced enzymes to degrade macromolecular carbohydrates [23,24,25].

### 3.2. Free and Bound TPC

Free, bound, and the sum of free and bound TPC of *Betaphycus gelatinum* substrate fermented by *L. brevis* for 0 and 60 h are shown in Figure 2. After fermentation by *L. brevis* for 60 h, free TPC of *Betaphycus gelatinum* substrate significantly increased from 328.94 ± 9.65 to 865.42 ± 29.29 μg GAE/g DW (*p* < 0.05), while bound TPC significantly decreased from 1389.80 ± 163.12 to 1004.90 ± 87.32 μg GAE/g DW (*p* < 0.05), with an insignificant variation in the sum of free and bound TPC (*p* > 0.05). Moreover, the ratio of free TPC to the sum of free and bound TPC increased from 19.14% to 46.27%, while the ratio of bound TPC to the sum of free and bound TPC decreased from 80.86% to 53.73%. Although few studies have focused on the effect of lactic acid bacteria fermentation on the bound polyphenols of seaweeds, lactic acid bacteria fermentation has been found to improve the content of free and/or total polyphenols in rice bran and *Momordica charantia* juice [28,33]. The decreased bound TPC may be due to microbial enzymes, which can break the linkage between bound polyphenols and macromolecular polysaccharides and subsequently release the corresponding polyphenols [7,10,28]. The elevated free TPC may be due to the release of bound polyphenols and the bioconversion of polyphenols [10,34].

### 3.3. Polyphenol Compound Identification

A total of 27 polyphenol compounds, including phenolic acids (7), flavonoids (18), and bromophenols (2), were identified (Table 1). 4-Hydroxybenzoic acid (peak 1) was tentatively identified by the loss of the carboxylic acid group (−44 Th) [35]. Ethyl vanillin was tentatively identified by virtue of the vanillin fragment (*m*/*z* 136.21) from the loss of an ethyl moiety [36]. *Trans*-cinnamic, caffeic, and ferulic acids showed the parent ions at *m*/*z* 146.95, 179.00, and 193.00, respectively, accompanied by their product ions at *m*/*z* 118.94 [M+H–CO]^+^ (peak 3), 135.0 [M–H–CO_2_]^−^ (peak 4), and 178.00 [M–H–CH_3_]^−^, and 149.00 [M–H–CO_2_]^−^ (peak 5), respectively [35,37]. Rosmarinic acid and 1,3-O-dicaffeoylquinic acid were identified by a dihydroxyphenyl-lactic acid fragment (*m*/*z* 196.96), a loss of H_2_O from the caffeic acid fragment (*m*/*z* 161.00) (peak 6), and a caffeoylquinic acid fragment (*m*/*z* 352.94) and quinic acid fragment (*m*/*z* 191.01) (peak 7), respectively [4,38].

Peak 8–10, with the parent ions [M–H]^−^ at *m*/*z* 304.98, 304.98, and 440.89 showed the [M–H–126]^−^ fragments (*m*/*z* 179.01, 179.01, and 288.99) corresponding to the loss of the C_6_H_6_O_3_ group, a loss of CO_2_ from the gallic acid fragment (*m*/*z* 124.98 and 124.97), and/or the gallic acid fragment (*m*/*z* 168.98), deduced as gallocatechin, epigallocatechin, and epicatechin gallate, respectively [39]. Gallocatechin and epigallocatechin were distinguished by different retention times. Peak 11 showed the parent ion [M–H]^−^ at *m*/*z* 301.08 with fragments at *m*/*z* 258.07 and 229.10, which were consistent with the reported fragments of ellagic acid [40]. Quercetin, morin, and myricetin were characterized by the typical retro Diel-Alder fragmentation of flavon-3-ols (*m*/*z* 179.00 and 151.00) [35,37,41]. Taxifolin displayed the [M–H–H_2_O]^−^ (*m*/*z* 284.98) and [M–H–C_6_H_6_O_3_]^−^ (*m*/*z* 176.91) fragments, which were consistent with MS data reported previously [42]. Hinokiflavone was identified by the parent ion [M+H]^+^ at *m*/*z* 537.00 and the fragment at *m*/*z* 284.00, which were in agreement with the MS data from a previous study [9]. Hyperoside, isoquercitrin, kaempferol-3-O-glucoside, cosemetin (apigenin-7-O-glucoside), vitexin (apigenin-8-C-glucoside), myricitrin (myricetin-3-O-rhamnoside), taxifolin-7-O-rhamnoside, and isorhamnetin-3-O-glucoside were distinguished by the loss of glycosides from their aglycones quercetin, quercetin, kaempferol, apigenin, myricetin, taxifolin, and isorhamnetin, respectively [35,43,44]. Cyanidin was distinguished by the parent ion [M+H]^+^ at *m*/*z* 286.91 and fragment ions [M+H–150]^+^ (*m*/*z* 137.01) and [M+H–150–CO]^+^ (*m*/*z* 109.05) [45].

Peak 26 displayed the parent ion [M–H]^−^ at *m*/*z* 250.74 and fragment ions at *m*/*z* 81.37 and 78.51, which was consistent with the MS data of 2,4-dibromophenol reported in a previous study [46]. The parent and product ions of peak 27 were observed at *m*/*z* 330.60 and 81.35, respectively, which were in agreement with the MS data of 2,4,6-tribromophenol reported previously [47].

Few studies have focused on the polyphenols of *Betaphycus gelatinum*. Polyphenol compounds have been identified in seaweed species from *Kappaphycus* and *Eucheuma* (which belong to the *Solieriaceae*) and other seaweeds. The presence of quercetin, catechin, and rutin was observed in the free extract of *Eucheuma cottonii* using HPLC [6], while free phlorotannin compounds (fucols, phlorethols, fucophloretols, fuhalols, and eckols) were not found in *Eucheuma denticulatum* using HPLC [5]. *Trans*-cinnamic acid, caffeic acid, ferulic acid, rosmarinic acid, ellagic acid, quercetin, myricetin, taxifolin, isoquercitrin, and vitexin were detected in *Kappaphycus alvarezii* by HPLC-MS [4]. Phenolic acids (including ethyl vanillin, 2-hydroxybenzoic acid, ferulic acid, rosmarinic acid, and *trans*-cinnamic acid), flavan-3-ols (gallocatechin and epigallocatechin), and cyanidin were found in the red seaweed *Asparagopsis taxiformis* using HPLC-MS [46]. Epicatechin gallate and hinokiflavone were abundant in the brown seaweeds *Anthophycus longifolius* and *Sargassum polycystum*, respectively [9,48]. Morin, quercetin, and myricetin were detected in the red seaweed *Acanthophora spicifera* by HPLC [49]. Taxifolin-O-rhamnoside, kaempferol-O-glucoside, myricetin-O-rhamnoside, apigenin-O-glucoside, quercetin-O-hexoside, and isorhamnetin-O-glucoside were identified in brown seaweeds by HPLC-MS [43], which may have been taxifolin-7-O-rhamnoside, kaempferol-3-O-glucoside, myricetin-3-O-rhamnoside (myricitrin), apigenin-7-O-glucoside (cosemetin), quercetin-3-O-galactoside (hyperoside), and isorhamnetin-3-O-glucoside, respectively, or their isomers. Simple bromophenols (including 2,4-dibromophenol and 2,4,6-tribromophenol) and complex bromophenols were found in several red seaweeds, including *Asparagopsis taxiformis*, *Polyopes lancifolia,* and *Rhodomela confervoides* [8,46,47]. These polyphenols were identified in *Betaphycus gelatinum* for the first time.

### 3.4. Polyphenol Compound Quantification

The individual polyphenol compounds in free and bound polyphenol extracts of *Betaphycus gelatinum* substrate fermented by *L. brevis* for 0 and 60 h were quantified by UPLC-QQQ-MS after being identified (Table 2). In *Betaphycus gelatinum* substrate without fermentation, 19 free and 24 bound polyphenols were found, while 23 free and 20 bound polyphenols were found after fermentation for 60 h. After *L. brevis* fermentation for 60 h, the contents of free ethyl vanillin, 1,3-O-dicaffeoylquinic acid, gallocatechin, ellagic acid, and 2,4,6-tribromophenol were increased by 103.37%, 32.64%, 106.12%, 61.99%, and 1447.86%, respectively (*p* < 0.05). Conversely, *L. brevis* fermentation led to significant decreases in the contents of free *trans*-cinnamic acid, epicatechin gallate, hinokiflavone, isorhamnetin-3-O-glucoside, and cyanidin, respectively (*p* < 0.05). Free ferulic acid, myricetin, kaempferol-3-O-glucoside, and vitexin were no longer present, while free caffeic acid, rosmarinic acid, epigallocatechin, hyperoside, isoquercitrin, taxifolin-7-O-rhamnoside, and 2,4-dibromophenol were newly found after fermentation. For bound polyphenol compounds, the contents of ethyl vanillin, *trans*-cinnamic acid, ferulic acid, rosmarinic acid, gallocatechin, epicatechin gallate, quercetin, hinokiflavone, myricitrin, isorhamnetin-3-O-glucoside, and cyanidin were decreased by 80.86%, 84.27%, 23.48%, 70.57%, 5.08%, 34.77%, 46.6%, 12.26%, 5.59%, 41.19%, and 74.86% after *L. brevis* fermentation, respectively (*p* < 0.05). Bound morin, isoquercitrin, taxifolin-7-O-rhamnoside, and 2,4-dibromophenol were no longer present after *L. brevis* fermentation. The contents of bound isoquercitrin and taxifolin-7-O-rhamnoside before fermentation were equal to those of free isoquercitrin and taxifolin-7-O-rhamnoside newly found after *L. brevis* fermentation, revealing the release of bound polyphenols to free polyphenols. *Trans*-cinnamic acid (40.92%), hinokiflavone (20.58%), ethyl vanillin (11.65%), and ferulic acid (9.66%) were the major free polyphenol components before fermentation, whereas rosmarinic acid (68.32%), *trans*-cinnamic acid (9.38%), ethyl vanillin (5.95%), hinokiflavone (4.78%), 2,4-dibromophenol (4.00%), and 2,4,6-tribromophenol (2.64%) were the major free polyphenol compositions after fermentation. *Trans*-cinnamic acid and rosmarinic acid were the main bound polyphenol components before fermentation, accounting for 62.65% and 29.39% of the sum of bound individual polyphenol content, respectively, whereas after fermentation, *trans*-cinnamic acid, rosmarinic acid, and hinokiflavone were the main bound polyphenol components, accounting for 41.96%, 36.82%, and 9.22% of the sum of bound individual polyphenol content. Bound *trans*-cinnamic acid (56.75%), bound rosmarinic acid (26.62%), and free *trans*-cinnamic acid (3.85%) were the main components before fermentation, while free rosmarinic acid (43.57%), bound *trans*-cinnamic acid (15.19%), bound rosmarinic acid (13.33%), and free *trans*-cinnamic acid (5.99%) were the main components after fermentation. Moreover, fermentation led to a significant increase in the sum of free individual polyphenol content from 206.48 ± 5.92 to 822.19 ± 26.70 μg/g DW, which resulted from the significant increases in the sum of free individual phenolic acid and bromophenol contents. Before fermentation, the sum of free individual phenolic acid, flavonoid, and bromophenol contents accounted for 64.18%, 35.15%, and 0.68% of the sum of free individual polyphenol content, while after fermentation the proportions were 84.34%, 9.03%, and 6.64%, respectively. Conversely, fermentation led to a significant decrease in the sum of bound individual polyphenol content from 1986.13 ± 8.32 to 466.51 ± 13.02 μg/g DW, which arose from the significant decreases in the sum of bound individual phenolic acid, flavonoid, and bromophenol contents. Before fermentation, the sum of bound individual phenolic acid, flavonoid, and bromophenol contents accounted for 93.77%, 4.87%, and 1.36% of the sum of bound individual polyphenol content, while after fermentation the proportions were 84.25%, 16.51%, and 0.24%, respectively. Furthermore, fermentation reduced the ratio of bound to free in the sum of individual polyphenol content from 9.62 to 0.52. Similarly, the ratio of bound TPC to free TPC declined from 4.23 to 1.16. These results revealed the significant effect of fermentation on the free and bound polyphenol profiles for the first time, including the release of bound polyphenols to free polyphenols and the conversion of the polyphenol structure [10,34].

Although there is a lack of information on the effect of lactic acid bacteria fermentation on the free and bound polyphenol profiles of *Betaphycus gelatinum* in previous studies, fermentation has been reported to change the free polyphenol profile in fruits and grains, accompanied by increased TPC, TFC, and/or antioxidant ability [13,14,15,50]. *Lactobacillus plantarum* fermentation significantly increases the content of free gallic acid in apple juice, which may be due to the hydrolysis of polymerized tannins and the subsequent formation of gallic acid [14]. The contents of free protocatechuic, isoferulic, and cinnamic acids in apple juice were increased by both *Saccharomyces cerevisiae* and *L. plantarum* fermentation, while the contents of free phloretin, rutin, and ellagic acid were decreased [14]. The content of free caffeic acid was increased by *S. cerevisiae* fermentation due to the release of bound caffeic acid or the degradation of polyphenols containing caffeic acid (such as 1,3-O-dicaffeoylquinic acid) [14]. *L. plantarum* fermentation led to a significant increase in the content of caffeic acid and then a subsequent considerable decrease [14]. During the fermentation of *Chenopodium quinoa* by *Lactobacillus*, the contents of free procyanidin B2, *p*-hydroxybenzaldehyde, protocatechuic acid, kaempferol, and rutin were increased, while those of catechin, procyanidin B1, and ferulic acid were decreased [50]. During fermentation of elderberry juices, caffeic and protocatechuic acids were consumed, while dihydrocaffeic acid was produced and anthocyanins changed in a strain-specific manner [51]. Usually, the increase in the content of polyphenol compounds in the early fermentation stage may be attributed to the release of bound polyphenols by enzymes produced by bacteria [14,52]. Lactic acid bacteria are able to produce glycosylases and polyphenolic esterases, including glucosidase, amylase, cellulase, chitinase, inulinase, phytase, xylanase, tannase, and esterase [52], which can break down the linkages between polyphenols and polysaccharides to release the bound polyphenols to the free form, hydrolyse glycosides to release aglycone, or hydrolyse the ester bonds in hydrolysable tannins and gallic acid esters [10,14,52]. Fermentation by lactic acid bacteria also contributed to the simple polyphenol conversion and the depolymerization of high-molecular-weight polyphenol compounds [16]. During the fermentation of elderberry juice by dairy strains, newly produced glycosides (including cyanidin-3-O-glucoside, quercetin-3-O-rutinoside, and cyanidin-3-O-sambubioside) were observed [51]. In this study, hyperoside and cosemetin were produced in accordance with the findings of a previous study [51]. Lactic acid bacteria were able to express glycosyltransferases, which were involved in the glycosylation of polyphenols to improve solubilization and detoxification [51]. The expression of glucansucrases in lactic acid bacteria was involved in the glycosylation of myricetin, quercetin, luteolin, and catechol [53]. Additionally, *Bifidobacteria* HN-3 fermentation produced deglycosylation, demethylation, hydrogenation, and esterification enzymes, which contributed to the conversion of the polyphenol structure [34]. These results were the first to show the free and bound polyphenol profiles of *Betaphycus gelatinum.* Moreover, this study also showed the variations in the free and bound polyphenol profiles of *Betaphycus gelatinum* induced by *L. brevis* fermentation for the first time.

### 3.5. Antioxidant Activity

The variations in the antioxidant activity of free and bound polyphenol extracts of *Betaphycus gelatinum* after *L. brevis* fermentation were evaluated by FRAP, DPPH, and ABTS assays (Figure 3). The ABTS scavenging assay is a radical scavenging process based on electron transfer. The antioxidant (polyphenol) transfers one electron to ABTS+ during the test procedure, reducing absorbance at 734 nm [54]. The DPPH scavenging assay is based on electron transfer as well. The antioxidant (polyphenol) gets one electron from DPPH during the test procedure, reducing color and absorbance at 517 nm [55]. The FRAP assay is based on antioxidants’ (polyphenols’) ability to convert iron (III) to iron (II). The reaction is then observed through an iron interaction with 2,4,6-tris(2-pyridyl)-s-triazine, which produces a violet-blue color and improves absorbance at 593 nm [56]. *L. brevis* fermentation for 60 h led to significant increases in the antioxidant activity of free polyphenol extracts of *Betaphycus gelatinum* substrate, as indicated by the remarkably increased FRAP value and the DPPH and ABTS radical scavenging capacities of free polyphenol extracts compared with those of *L. brevis* fermentation for 0 h. Conversely, *L. brevis* fermentation for 60 h brought about a significant decrease in the antioxidant activity of bound polyphenol extracts of *Betaphycus gelatinum* substrate, as indicated by the lower DPPH and ABTS radical scavenging capacities of bound polyphenol extracts compared with those of *L. brevis* fermentation for 0 h. There was no significant difference in the sum of the FRAP values of free and bound polyphenol extracts between 0 and 60 h fermentation. Generally, the trend in variation in the antioxidant activity of free polyphenol extracts during fermentation was consistent with that of previous studies [14,15,17,18,28,57]. The antioxidant abilities of free polyphenol extracts of apple juice, watermelon juice, rice bran, *Sargassum* sp., and Irish brown seaweeds after lactic acid bacteria fermentation were higher than those before fermentation [14,15,17,18,28]. Similar results were also found in the antioxidant activity of free *Citrus aurantium* flower extracts fermented by *L. brevis* [57]. In a previous study, it was found that the enzymes produced by lactic acid bacteria during fermentation released the bound polyphenols, thus increasing the content and antioxidant ability of the free polyphenols [58].

### 3.6. Correlation Analysis between TPC and Antioxidant Activity

A heatmap of the correlation among free TPC, bound TPC, the sum of free and bound TPC, and antioxidant activity is shown in Figure 4, and correlation coefficient among them is shown in Appendix A. There was a positive correlation between free antioxidant activity and free TPC, as indicated by Pearson’s r values of 0.994 (*p* ≤ 0.001), 0.981 (*p* ≤ 0.001), and 0.968 (*p* ≤ 0.01) between free TPC and the FRAP value and the DPPH and ABTS radical scavenging capacities of free polyphenol extracts, respectively. Similarly, a correlation between bound TPC and the FRAP value and the DPPH and ABTS radical scavenging capacities of bound polyphenol extracts (Pearson’s r = 0.816 [*p* ≤ 0.05], 0.983 [*p* ≤ 0.001], and 0.978 [*p* ≤ 0.01]) revealed a positive correlation between bound antioxidant activity and bound TPC. A positive correlation between the TPC and antioxidant activity was shown in previous studies [28,57]. Moreover, there was a negative correlation between free TPC and bound TPC (Pearson’s r = −0.849 [*p* ≤ 0.05]), which was consistent with the changing trend of free and bound TPC after fermentation and supported the view that fermentation led to the release of bound polyphenols into free polyphenols. Furthermore, a negative correlation between free TPC and DPPH and ABTS radical scavenging capacities of bound polyphenol extracts (Pearson’s r = −0.887 [*p* ≤ 0.05] and −0.931 [*p* ≤ 0.01]) revealed a negative correlation between bound antioxidant activity and free TPC. A negative correlation between the ABTS radical scavenging capacities of bound polyphenol extracts and the FRAP value and the DPPH and ABTS radical scavenging capacities of free polyphenol extracts (Pearson’s r = −0.886, −0.859, and −0.812 [*p* ≤ 0.05]) revealed a negative correlation between bound antioxidant activity and free antioxidant activity. The negative correlation between bound antioxidant activity and free TPC and antioxidant activity also indirectly supported the view that fermentation led to the release of bound polyphenols into free polyphenols [10,34].

## 4. Conclusions

*Betaphycus gelatinum* pretreated by ultrasound-assisted mild acid hydrolysis can be fermented by *L. brevis*, as indicated by the increased viable colony number, the decreased pH, and the dynamically changed reducing sugar content. Moreover, *L. brevis* fermentation for 60 h led to significant increases in free TPC and antioxidant activity and significant decreases in bound TPC and antioxidant activity. Free and bound TPC were positively correlated with free and bound antioxidant activity, respectively. Bound TPC and antioxidant activity were negatively correlated with free TPC and antioxidant activity. Furthermore, 19 free and 24 bound polyphenols were identified by UPLC-QQQ-MS before fermentation, while 23 free and 20 bound polyphenols were identified after fermentation for 60 h. Bound *trans*-cinnamic acid, bound rosmarinic acid, and free *trans*-cinnamic acid were the main components before fermentation, while free rosmarinic acid, bound *trans*-cinnamic acid, bound rosmarinic acid, and free *trans*-cinnamic acid were the main components after fermentation. These results indicated that *Betaphycus gelatinum,* rich in polyphenols, can be processed into functional foods.

## Figures and Tables

**Figure 1 foods-12-03334-f001:**
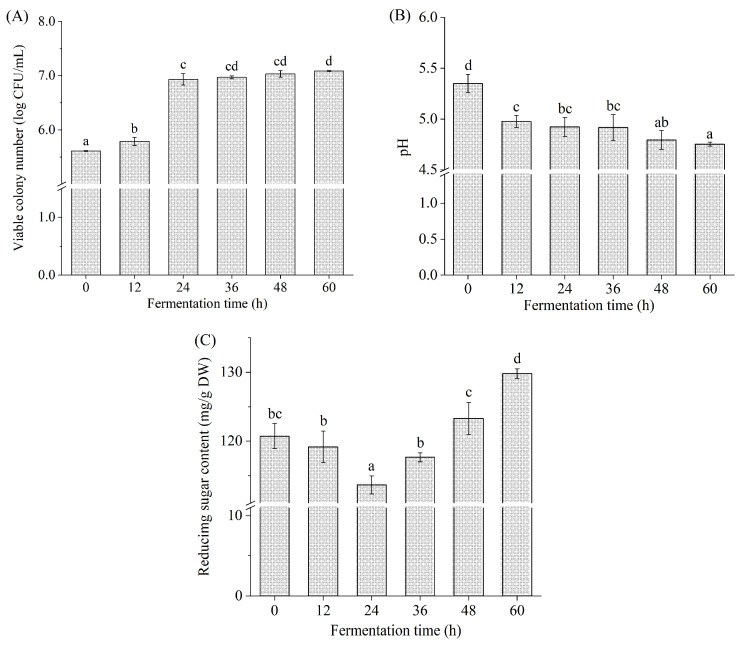
Viable colony number (**A**), pH (**B**), and reducing sugar content (**C**) during *L. brevis* fermentation. Different lowercase letters represent significant differences (*p* < 0.05).

**Figure 2 foods-12-03334-f002:**
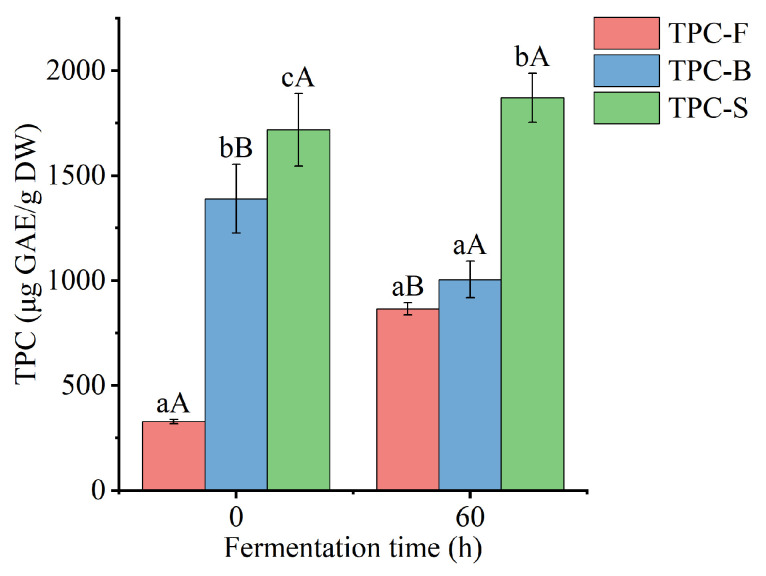
Free, bound, and the sum of free and bound TPC of *Betaphycus gelatinum* substrate fermented by *L. brevis* for 0 h and 60 h. TPC-F: free TPC; TPC-B: bound TPC; TPC-S: the sum of free and bound TPC; Different lowercase letters within the group represent significant differences (*p* < 0.05). Different capital letters between 0 h and 60 h represent significant differences (*p* < 0.05).

**Figure 3 foods-12-03334-f003:**
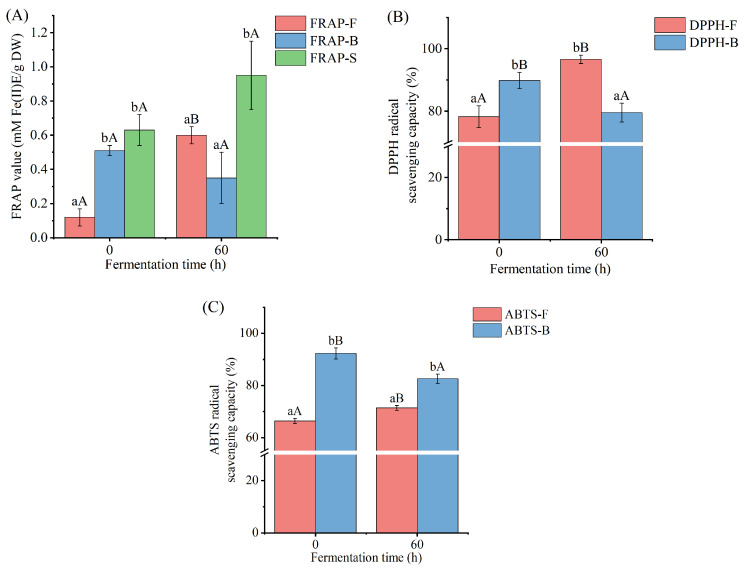
FRAP value (**A**), DPPH radical scavenging capacity (**B**), and ABTS radical scavenging capacity (**C**) of free and bound polyphenol extracts of *Betaphycus gelatinum* substrate fermented by *L. brevis* for 0 h and 60 h. FRAP–F: FRAP of free polyphenol extracts; FRAP–B: FRAP of bound polyphenol extracts; DPPH–F: DPPH radical scavenging capacities of free polyphenol extracts; DPPH–B: DPPH radical scavenging capacities of bound polyphenol extracts; ABTS–F: ABTS radical scavenging capacities of free polyphenol extracts; ABTS–B: ABTS radical scavenging capacities of bound polyphenol extracts. Different lowercase letters within the group represent significant differences (*p* < 0.05). Different uppercase letters between 0 h and 60 h represent significant differences (*p* < 0.05).

**Figure 4 foods-12-03334-f004:**
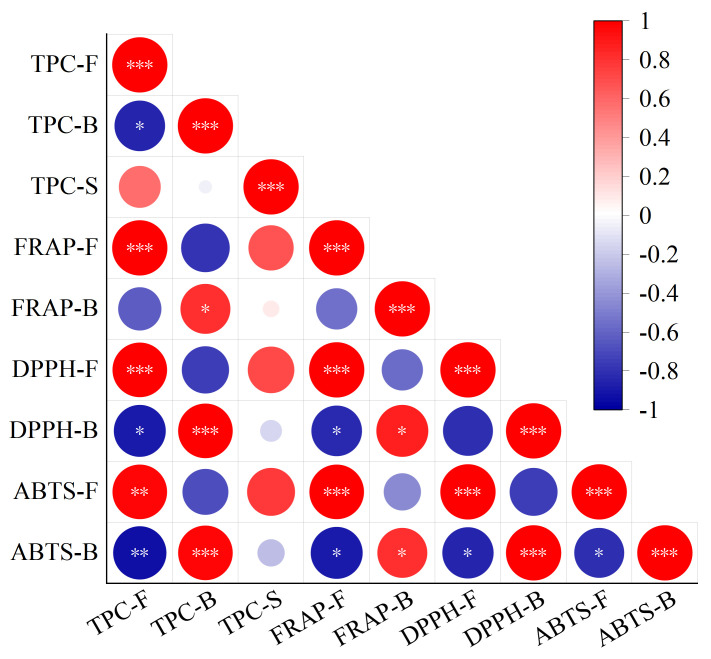
Heatmap of the correlation among the free, bound, and sum of free and bound TPC and antioxidant activity of *Betaphycus gelatinum* substrate fermented by *L. brevis* for 0 h and 60 h. TPC: total phenolic content; TPC–F: free TPC; TPC–B: bound TPC; TPC–S: the sum of free and bound TPC; FRAP–F: FRAP of free polyphenol extracts; FRAP–B: FRAP of bound polyphenol extracts; DPPH–F: DPPH radical scavenging capacities of free polyphenol extracts; DPPH–B: DPPH radical scavenging capacities of bound polyphenol extracts; ABTS–F: ABTS radical scavenging capacities of free polyphenol extracts; ABTS–B: ABTS radical scavenging capacities of bound polyphenol extracts. * Correlation is significant at the 0.05 level; ** Correlation is significant at the 0.01 level; *** Correlation is significant at the 0.001 level.

**Table 1 foods-12-03334-t001:** The identification of individual polyphenol compounds in the free and bound polyphenol extracts of *Betaphycus gelatinum* substrate fermented by *L. brevis* for 0 h and 60 h.

Peak No.	Tentative Assignment	ESI Model	Parent Ions	Fragment Ions	Reference
1	4-hydroxybenzoic acid	−	137.00	93.00	[35], standard
2	ethyl vanillin	−	164.95	136.21, 92.05	[36], standard
3	*trans*-cinnamic acid	+	146.95	118.94, 77.01, 40.07	[35], standard
4	caffeic acid	−	179.00	135.00, 79.00	[37], standard
5	ferulic acid	−	193.00	178.00, 149.00, 134.00	[37], standard
6	rosmarinic acid	+	358.96	196.96, 161.00	[4], standard
7	1,3-O-dicaffeoylquinic acid	−	514.91	352.94, 191.01	[38], standard
8	gallocatechin	−	304.98	179.01, 124.98	[39], standard
9	epigallocatechin	−	304.98	179.01, 124.98	[39], standard
10	epicatechin gallate	−	440.89	288.99, 168.98, 124.97	[39], standard
11	ellagic acid	−	301.08	258.07, 229.10	[40], standard
12	quercetin	−	301.01	178.99, 150.99	[35], standard
13	morin	−	300.83	150.99, 124.98 106.97	[41], standard
14	myricetin	−	317.00	178.95, 150.99	[37], standard
15	taxifolin	−	302.83	284.98, 176.91, 124.98	[42], standard
16	hinokiflavone	+	537.00	417.00, 284.00	[9], standard
17	hyperoside	−	463.00	300.90, 270.90, 254.90	[43], standard
18	isoquercitrin	−	447.00	301.00, 179.00, 151.00	[43], standard
19	kaempferol-3-O-glucoside	−	446.90	285.06, 254.94, 226.98	[43], standard
20	cosemetin	−	430.90	268.90	[43], standard
21	vitexin	−	431.00	310.90, 340.80, 282.90	[44], standard
22	myricitrin	−	462.90	316.99	[43], standard
23	taxifolin-7-O-rhamnoside	−	448.85	302.92, 284.91, 124.98	[43], standard
24	isorhamnetin-3-O-glucoside	−	476.86	313.96, 242.84	[43], standard
25	cyanidin	+	286.91	109.05, 137.01	[45], standard
26	2,4-dibromophenol	−	250.74	168.70, 81.37, 78.51	[46], standard
27	2,4,6-tribromophenol	−	330.60	81.35	[47], standard

**Table 2 foods-12-03334-t002:** The content of individual polyphenol compounds in the free and bound polyphenol extracts of *Betaphycus gelatinum* substrate fermented by *L. brevis* for 0 h and 60 h.

	Free Polyphenol Content (μg/g DW)		Bound Polyphenol Content (μg/g DW)	
	0 h	60 h	0 h	60 h
4-hydroxybenzoic acid	1.64 ± 0.02 a	1.61 ± 0.02 a	1.84 ± 0.04 A	1.79 ± 0.03 A
ethyl vanillin	24.05 ± 0.70 a	48.91 ± 1.51 b	11.65 ± 0.31 B	2.23 ± 0.03 A
*trans*-cinnamic acid	84.49 ± 3.47 b	77.16 ± 0.47 a	1244.25 ± 2.66 B	195.74 ± 5.51 A
caffeic acid	nd	1.60 ± 0.01 a	1.63 ± 0.02 A	1.67 ± 0.03 A
ferulic acid	19.94 ± 0.56 a	nd	16.74 ± 0.67 B	12.81 ± 0.32 A
rosmarinic acid	nd	560.94 ± 21.56 a	583.75 ± 1.25 B	171.77 ± 4.90 A
1,3-O-dicaffeoylquinic acid	2.39 ± 0.002 a	3.17 ± 0.16 b	2.49 ± 0.05 A	2.38 ± 0.08 A
*number of phenolic acids*	*5*	*6*	*7*	*7*
*sum of individul phenolic acid content*	*132.51 ± 4.75 a*	*693.40 ± 23.72 b*	*1862.34 ± 4.99 B*	*388.38 ± 10.89 A*
gallocatechin	4.25 ± 0.04 a	8.76 ± 0.26 b	9.44 ± 0.27 B	8.96 ± 0.08 A
epigallocatechin	nd	4.73 ± 0.19 a	4.23 ± 0.06 A	4.38 ± 0.20 A
epicatechin gallate	3.51 ± 0.15 b	2.39 ± 0.08 a	3.71 ± 0.01 B	2.42 ± 0.08 A
ellagic acid	2.71 ± 0.05 a	4.39 ± 0.12 b	4.63 ± 0.15 A	4.53 ± 0.03 A
quercetin	1.62 ± 0.05 a	1.60 ± 0.05 a	3.09 ± 0.01 B	1.65 ± 0.004 A
morin	1.19 ± 0.03 a	1.15 ± 0.002 a	1.17 ± 0.004 A	nd
myricetin	2.28 ± 0.07 a	nd	nd	nd
taxifolin	1.13 ± 0.01 a	1.14 ± 0.002 a	1.13 ± 0.01 A	1.13 ± 0.004 A
hinokiflavone	42.50 ± 0.42 b	39.33 ± 0.54 a	49.03 ± 1.40 B	43.02 ± 1.35 A
hyperoside	nd	0.56 ± 0.002 a	0.58 ± 0.01 A	0.60 ± 0.03 A
isoquercitrin	nd	0.67 ± 0.01 a	0.68 ± 0.02 A	nd
kaempferol-3-O-glucoside	1.18 ± 0.01 a	nd	nd	nd
cosemetin	nd	0.81 ± 0.002 a	nd	nd
vitexin	3.06 ± 0.10 a	nd	3.46 ± 0.10 A	3.31 ± 0.15 A
myricitrin	3.24 ± 0.09 a	3.22 ± 0.09 a	3.40 ± 0.04 B	3.21 ± 0.08 A
taxifolin-7-O-rhamnoside	nd	1.50 ± 0.002 a	1.50 ± 0.002 A	nd
isorhamnetin-3-O-glucoside	3.44 ± 0.004 b	1.97 ± 0.08 a	3.35 ± 0.09 B	1.97 ± 0.09 A
cyanidin	2.47 ± 0.09 b	2.01 ± 0.06 a	7.28 ± 0.22 B	1.83 ± 0.004 A
*number of flavonoids*	*13*	*15*	*15*	*12*
*sum of individual flavonoid content*	*72.58 ± 1.11 a*	*74.22 ± 1.48 a*	*96.68 ± 2.39 B*	*77.00 ± 2.09 A*
2,4-dibromophenol	nd	32.91 ± 0.89 a	24.83 ± 0.88 A	nd
2,4,6-tribromophenol	1.40 ± 0.06 a	21.67 ± 0.62 b	2.27 ± 0.06 B	1.13 ± 0.04 A
*number of bromophenols*	*1*	*2*	*2*	*1*
*sum of individual bromophenol content*	*1.40 ± 0.06 a*	*54.57 ± 1.50 b*	*27.11 ± 0.94 B*	*1.13 ± 0.04 A*
*number of polyphenols*	*19*	*23*	*24*	*20*
*sum of individual polyphenol content*	*206.48 ± 5.92 a*	*822.19 ± 26.70 b*	*1986.13 ± 8.32 B*	*466.51 ± 13.02 A*

Different lowercase letters indicate significant differences between free polyphenol content (*p* < 0.05). Different uppercase letters indicate significant differences between bound polyphenol content (*p* < 0.05).

## Data Availability

Data are contained within the article.

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
