# Peer review of "Fermentation of *Betaphycus gelatinum* Using *Lactobacillus brevis*: Growth of Probiotics, Total Polyphenol Content, Polyphenol Profile, and Antioxidant Capacity"

_foods, 2023, doi:10.3390/foods12183334_

Round 1
Reviewer 1 Report
I am not the expert for fermentation technology, so I would not comment it in details. However, concerning determination of phenolic compounds, there are a lot of inadequate explanations or misleading information. It is not clear how the authors determined phenolic content and how further identification and quantification of individual compounds was performed. Since half of the manuscript relies on this data, the authors need to provide more information of these methods and results. It is not acceptable in this form. More specific comments are given bellow. English language should be also improved throughout the whole manuscript.
Abstract: should be reorganised
The first sentence: rephrase it
Line 21: replace the word “counts” with more appropriate one
Line 29: The sentence cannot start with the number
Lines 30-33: the sentence is too long. Divide it in shorter sentences.
Introduction
Line 39: there should not be written (B. gelatinum) as an abbreviation, because it is normative that every next usage of Latin name of the specie would be written that way.
Line 39: avoid “excellent”
Line 40: what are Betaphycus, Solieriaceae? Family or? It should be written.
Line 41: breeding seaweeds??
Line 60: the term “rich nutrients” cannot be used like this
There is nowhere clearly given explanation of importance of fermentation process in this research and why it was performed.
Lines 94-95: the last sentence is not appropriate, rephrase it, or wreplace it with more appropriate explanation of the purpose of the work.
Materials and methods
Lines 108-109: provide the whole name for MRS (MRS medium and MRS broth)
Paragraph 2.6.: provide more information on used methods
Paragraph 2.7.: there is explained only extraction of bound polyphenols. How did you determined free phenolics? How did you determine total phenolics? Was it a simple addition of free and bound phenolics or…? It need to be clearly stated in the text, both here and in the results.
Paragraph 2.8.: provide more information on used method
Line 160: which polyphenol standards? Specify each compound (here and in “Chemicals and reagents” paragraph). Provide calibration curves parameters for each compound.
Results
Lines 192-193: beneficial effect on consumer’s health – you cannot state this from the observed results
Line 196-197: rephrase the sentence to provide adequate meaning
Line 226: how total phenolic content was calculated?
Abbreviation TPC means “total phenolic content”, so you cannot say “total TPC”
Table 1. The identification of individual polyphenol compounds is not performed properly. Some parent and fragment ions does not match the reference. For instance, the authors refer to the reference 37 for the majority of the compounds, and some of these compounds are not found in reference 37 (taxifolin…)
If you had standard compounds, why did you rely only to literature?
Table 2? How did you quantify each compound? Which standards you used?
Line 402: again, it is not clear how you get free and bound polyphenol extracts
Paragraphs 3.5 and 3.6: there cannot be used terms “free” or “bound” FRAP, DPPH, ABTS values
Conclusion: should be reorganised. The sentences are too long. It looks more like listing results than providing a real conclusion.
English language should be improved throughout the whole manuscript.
Reviewer 2 Report
Main question addressed by the research: In the present study, B. gelatinum pretreated by ultrasound-assisted mild acid hydrolysis was fermented by L. brevis. Firstly, the growth of L. brevis was monitored. Secondly, TPC, polyphenol profile, and antioxidant activity in both free and bound polyphenol extracts after fermentation for 0 h and 60 h were measured. Finally, the correlation among free, bound and total TPC and antioxidant activity was analyzed. This research provides useful information for the comprehensive utilization of B. gelatinum.
However, the authors did not clearly point out what was new in their study.
The topic in this study is original, relevant and specific in this field of research.
However, the authors should highlight this in more detail, in the manuscript itself: originality, relevance, specificity, novelty..., of their research topics in this area.
This submitted manuscript is interesting and deals with a current and innovative topic in the field of biotechnology and/or biochemical engineering.
However, I have a suggestion to improve the manuscript in the Introduction so that it contains more information about edible seaweeds and others with a similar purpose, which would further emphasize the importance of the topic of this manuscript in relation to similar previously published and now available materials in the literature.
The characteristics of the used equipment, chemicals, reagents and consumables must be supplemented, in a way that has already become common in "scientific writing". Also, techniques (especially instrumental testing) always require a detailed description of the equipment used, characteristics, working conditions, standard chemicals used, etc., and this is already a routine and usual situation in the scientific presentation of experimental results in manuscripts. The authors should rearrange and supplement it. For example, in sections 2.6 (F meter used, buffer used, etc.), in sections 2.8 and 2.10, for spectrophotometric techniques, spectrophotometers used, etc., equipment data, standard chemicals used, calibration standards, etc., are missing, if applicable.
Part of the manuscript Results and discussion are in accordance with the performed experimental plan, however, some improvements are needed.
The results are presented using two tables and four figures.
There are some unclear elements and shortcomings in the tables and figures.
For the results shown in Figures 2 and 3 (A, B and C), * is not an adequate option to show between the results significant differences between values at 0 h and 60 h of fermentation (p < 0.05). The histograms should be displayed differently. And for statistically significant differences, between o h and 60 h of fermentation (p < 0.05), choose for example uppercase (A, B, ...).
In Figure 4, it is not clearly indicated what the "asterisks" represent, one (*), two (**) and three (***).
The present conclusion generally follows the discussion of the results, but what hasa been said is relevant. However, it would have to be improved, by one or more "main" conclusions. For example, how the obtained results can provide scientific information for processing B. gelatinum, as the authors claim in the Abstract, and similar "general" conclusions if there are more, it would be good to "incorporate" them into the Conclusion.
The listed references are appropriate. Considering the above, it will be necessary for the authors to improve and supplement them.
Although I am not an expert in English, I think that the English language in this manuscript must be improved.
Reviewer 3 Report
Dear authors, the information included in this manuscript is novel, however, it is necessary to review the following comments:
Line 18: for food industry?
Line 29: modify…. In total, 19 free…
Line 41: could include statistical information on the production of carrageenans that are produced from B. gelatinum?
Line 46: use [4–6] instead of [4,5,6]
Line 59: use [12–14] instead of [12-14]
Line 61: use [12–15] instead of [12-15]
Line 65: use [13–15] instead of [13-15]
Line 69: use [13–15] instead of [13-15]
Line 101: were samples packed under vacuum?
Line 116: insert space…75 °C
Line 117: add information about the sterilizer equipment (model, trademark, country)
Line 118: insert space…121 °C
Line 122: insert space…37 °C
Line 124: add information about the centrifuge equipment (model, trademark, country)
Line 128: insert space…37 °C
Line 138: as described previously [9], with some modifications.
Line 141: insert space…4 °C
Line 143: insert space…45 °C
Line 144: insert space…-18 °C
Note: insert spaces between the number and temperature symbol through the manuscript
Line 163: insert space…. 4.5 kV
Line 164: (N2)
Line 169: method reported previously [16].
Line 193: that the fermented B. gelatinum had a beneficial effect on consumer’s health [reference?].
Line 194: in figures images use a parenthesis for each letter (A), (B) and (C)
Line 200: S. japonica
Line 224: use [20–22] instead of [20-22]
Line 238: [7,10,28].
Line 252: [31,33].
Line 266: reported previously [37].
Line 279: reported previously [40].
Line 310: insert space… 60 h. After
Line 329,338,339: trans-cinnamic
Line 371: use italic text format for scientific names
Line 401: in the discussion of this section, explain the antioxidant mechanism by which the methos used work.
Line 413: [14,15,17,18,28,49].
Line 416: [14,15,17,18,28].
Line 417: [49]. In a previous study it was found
Line 477: use [15–23] instead of [15-23]
Line 479: use [599–605] instead of [599-605]
Line 481: insert the correct information 145–146 or 135–144.
Line 486: 16–30
Line 489: 376–382
Line 491: 626–659
Line 496: 968–981
Line 498: 347–362
Line 515: 346–355
Line 517: 96–107 (note: modify trough the reference section)
Round 2
Reviewer 1 Report
The authors corrected the manuscript properly. The quality is much better now. Still, there are some minor things that should be corrected.
Line 173: delete "extracts"
Line 189: add "total phenolic content" before (TPC)
Line 531: increased
Supplementary material: Table 2, statistically significant correlations should be marked in bold. It is usual to delete the values above or under the middle part of the table (where all the values are 1).
The English language is appropriate.
Author Response
请参阅附件。

Reviewer 2 Report
The authors have taken into account the comments previously made by the reviewers.
They subsequently submitted a significantly improved manuscript.
I consider it now adequate for acceptance for publication.
Author Response
It is an honor to have your recognition of this work, and thank you very much for your time involved in reviewing the manuscript.